# Anatomical analysis of antebrachial cutaneous nerve distribution pattern and its clinical implications for sensory reconstruction

Hui Li[1], Weiwei Zhu[1], Shouwen Wu[1], Zairong Wei[2], Shengbo Yang[1]*

**1** Department of Anatomy, Zunyi Medical University, Zunyi, Guizhou, China, **2** Department of Plastic Surgery, The First Affiliated Hospital of Zunyi Medical University, Zunyi, Guizhou, China

* yangshengbo8205486@163.com

**Data Availability Statement:** Data are shown in Table 1.

**Funding:** This work was supported by the National Natural Science Foundation of China (No.

## Abstract

This study aimed to reveal the distribution pattern of antebrachial cutaneous nerves and provide a morphological basis for sensory reconstruction during flap transplantation. Forearm specimens containing skin and subcutaneous fat were obtained from 24 upper extremities of 12 adult cadavers. Cutaneous nerves were visualized using modified Sihler's staining. Then the data was used to show the distribution pattern and innervation area of the forearm cutaneous nerve. The anterior branch of lateral antebrachial cutaneous nerve innervates 26% of the medial anterior forearm; the posterior branch innervates 38.21% of the lateral anterior forearm and 24.46% of the lateral posterior forearm. The anterior branch of medial antebrachial cutaneous nerve innervates the medial aspect of the forearm covering 27.67% of the anterior region; the posterior branch the lateral part of the forearm covering 7.67% and 34.75% of the anterior and posterior regions, respectively. The posterior antebrachial cutaneous nerve covers 41.04% of the posterior forearm. Coaptations were found between the branches of these cutaneous nerves. The relatively dense secondary nerve branches were found in the middle 1/3 of the lateral anterior forearm and the middle 1/3 of the medial posterior forearm. The relatively dense tertiary nerve branches were the middle 1/3 and lower 1/3 of the medial anterior forearm. The intradermal nerve branches were the relatively dense in the middle 1/3 of the medial anterior and lateral posterior forearm. The middle 1/3 of the medial and lateral forearm had the relatively dense total nerve branches. These results can be used sensory matching while designing forearm flaps for reconstruction surgeries to obtain improved recovery of sensory.

## Introduction

Skin sensation in the forearm is innervated by the lateral antebrachial cutaneous nerve (LACN), the medial antebrachial cutaneous nerve (MACN), and the posterior antebrachial cutaneous nerve (PACN). Many gross anatomy reports on the origin, distribution, position to

31660294 to SY). The funder had no role in study design, data collection and analysis, decision to publish, or preparation of the manuscript.

**Competing interests:** The authors have declared that no competing interests exist.

veins, major branches, and dominant distribution regions of these nerves exist [1–6]. These studies are valuable, but they have limitations as the manipulation of nerves within soft tissues during dissection can alter its course and distribution. Most studies revealed only thick nerve branches without discussing the final distributions of the small nerves and locations of intensively distributed regions. Some studies have revealed the distribution density of fine nerves in some forearm skin regions by histological methods, but these studies lack the overview of the entire forearm's nerve distribution pattern [7–9].

Understanding the detailed distribution pattern of cutaneous nerves in the forearm is helpful for the sensory recovery of the recipient region when using forearm flaps for transplantation, which can be used to repair soft tissue defects, especially in sensitive regions, including the hands, mouth, and penis [10–12]. Previously, during flap transplantation, clinicians focused on maintaining blood vessels in the grafts to ensure graft survival, with limited concern for sensory reconstruction, causing lack of sensation in the donor grafts at the recipient site. Recently, some clinicians are actively designing and using vascular-neurotrophic flaps to reconstruct sensory functions at the recipient site [13]. However, two issues are associated with this approach: cutaneous nerve trunk removal can result in skin sensory deficits at the donor site, and the nerve density of the selected donor site does not necessarily match the needs of the recipient site. If an area innervated by a primary or secondary nerve branch with dense branches can be designed as a transplant object, it can reduce the sensory defect in the donor site, and also meet the needs of sensory reconstruction in the recipient site. However, at present, it is limited to not knowing the distribution pattern of the cutaneous nerve. Additionally, the distribution boundaries of forearm cutaneous nerves are still unclear when used for forearm vascular-neurotrophic flap transplantation. To this end, Rhee et al. used mechanical stimulation to detect the extent of LACN innervations and demonstrated a larger range than described previously through gross anatomy dissections [14–16]. Therefore, detailed studies of the cutaneous nerve distribution pattern of the forearm are necessary.

Sihler's staining can clearly display the entire intramuscular nerve distribution pattern between gross anatomy and microscopic details [17, 18]. Recently, using this method, some researchers have successfully demonstrated the cutaneous nerve distribution patterns of the occipital region and trigeminal nerves [19, 20]. This study assessed the distribution pattern of the forearm cutaneous nerves using Sihler's staining to provide a morphological basis for sensory reconstruction during skin flap transplantation.

## Materials and methods

### Specimens and ethics

Twelve donated adult cadavers, 8 men and 4 women, without history of diabetes mellitus, neurological deficits, or pathological skin conditions, were fixed with formalin. The causes of death of these donors are cancer, heart disease, and cerebrovascular accidents. None of the cadaver donors were from a vulnerable population and all donors or their next of kin provided signed written consent forms before being accepted for use in this project. Zunyi Medical University is authorized by Zunyi Red Cross Society as one of the registered institutions to accept willed body donation. The research protocol was pre-approved by the Ethics Commission of the Zunyi Medical University (approval #2016-1-006).

### Gross anatomy, marking, and measurement

A longitudinal incision was made in the lateral forearm between the lateral epicondyle of humerus and the tip of radial styloid process. Two transverse incisions were made between the medial and lateral epicondyles of humerus and between the tip of radial styloid process and

the tip of ulnar styloid process. The forearm skin containing subcutaneous fat was removed close to the muscle surface. Two segments of fishing line were sutured to each specimen, one at the proximal end approximately two finger-width below the lateral epicondyle incision, and another at the base of the styloid process of radius. The sutures marked the boundary between anterior and posterior forearm. Specimens were measured with Vernier calipers for length, width, and thickness.

### Forearm regions

The following lines and regions were established for a detailed analysis of the forearm cutaneous nerve distribution: [a]: lateral epicondyle of humerus; [a']: at two finger width below [a]; [b]: medial epicondyle of humerus; [b']: at two finger width below [b]; [c]: tip of the styloid process of radius; [c']: proximal base of the styloid process of radius; [d]: tip of styloid process of ulna, and [d']: proximal base of the styloid process of ulna. In the anterior forearm, the midpoint of the line connecting [a'] and [b'] was [e], and the midpoint between [c'] and [d'] was [e']. In posterior forearm, [f] and [f'] were designated in the same way as [e] and [e']. The anterior forearm was divided into medial and lateral regions by line [e]-[e'] and similarly the posterior forearm by line [f]-[f']. Then, the lines connecting [a'] and [c'] as well as [b'] and [d'] were further divided into three equal length segments, i.e. upper, middle, and lower. Hence, the anterior forearm was divided into 6 regions: medial upper, medial middle, medial lower, lateral upper, lateral middle, and lateral lower (1/3 each). The posterior forearm was divided into 6 regions in the same manner: medial upper, medial middle, medial lower, lateral upper, lateral middle, and lateral lower (1/3 each) (Fig 1).

### Scheme of the modified Sihler's staining

First, the specimens were degreased in absolute ethanol for 3 days, followed by hydrolyzation in 0.25% collagenase for 3 days. Finally, the specimens were subjected to Sihler's intramuscular nerve staining as described in our previous study [17, 18], briefly summarized as follows: specimens were immersed in 0.2% hydrogen peroxide + 3% potassium hydroxide solution for 4–5 weeks, then in Sihler's I solution (1 part glacial acetic acid, 2 parts glycerin, 12 parts 1% hydrated trichloroacetaldehyde) to decalcify for 4–5 weeks; specimens were then stained in Sihler's II solution for 4 weeks (1 part Ehrlich hematoxylin solution, 2 parts glycerin, 12 parts 1% hydrated trichloroacetaldehyde); specimens were re-immersed into Sihler's I solution for 2-10h; neutralized in 0.05% lithium carbonate solution for 2h; and treated with gradient glycerol (40%, 60%, 80% and 100%) for 1 week.

### Observation and measurement after staining

The stained specimens were placed on alighted box to observe cutaneous nerve distribution, including the distribution of primary, secondary, and tertiary nerve branches and the nerve-dense regions. Fishing lines were placed to connect the above-mentioned points. Regions were measured and photographed. Additional subcutaneous fat tissue was trimmed to expose intradermal nerve branches, with subsequent measurement, photographing, and drawing. It was necessary to remeasure the length and width of the specimens with Vernier calipers and calculate the scaling coefficient of the specimens' area: Coefficient = specimen's area after staining / specimen's area before staining as the specimens shrank during decalcification.

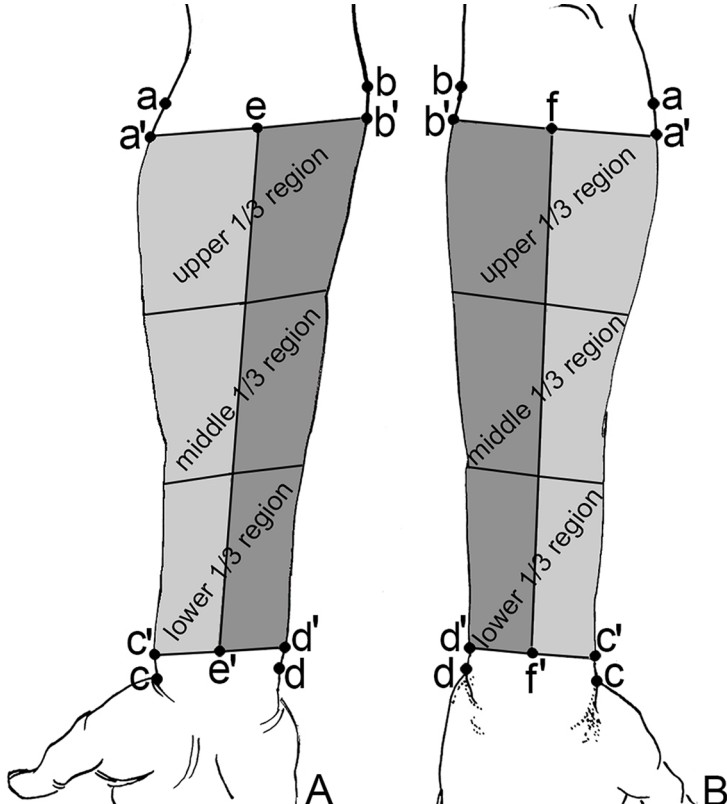

**Fig 1. Sketch map of forearm division.** (A) Anterior forearm. (B) Posterior forearm.

### Measurement of nerve branch density

Pictures were analyzed using Adobe Photoshop 13.0 software. Vertical and horizontal reference line tools were used to set the rectangular box size of 1×1cm, from top to bottom, then drag the reference line from left to right. Thereafter, the density of secondary, tertiary, and intradermal nerve branches in each region was counted. Nerve branch density in the region = (total number of nerve branches in the region / area of specimens in the region) × scaling coefficient. Finally, the sum of all nerve branches was done to calculate density.

### Statistical analyses

Data were analysed using the Statistical Package for the Social Sciences (SPSS) V.17.0 (SPSS Inc, Chicago, Illinois, USA). One-way ANOVA was used to compare data on thickness, area and nerve branch densities among different regions, and the Games-Howell multiple comparisons test was used as a post-hoc test. The comparison between two sides was done using paired $t$ test. with statistical significance as $P < 0.05$.

## Results

### Gross anatomy observation

At the level between the medial and lateral epicondylar connection in the humerus, 95.83% (23/24) and 4.17% (1/24) of the LACNs were divided into 2 (anterior and posterior) and 3 primary nerve branches, respectively. Furthermore, 95.83% (23/24) of the MACNs were divided into 2 primary nerve branches (anterior and posterior), and 4.17% (1/24) had only one trunk.

Moreover, 95.83% (23/24) of the PACNs had only one trunk, and 4.17% (1/24) were divided into medial and lateral primary nerve branches. Anatomical analyses also revealed that the nerve branches directly penetrated from the muscles to the skin in all regions of the forearm.

### Distribution pattern of cutaneous nerve

Stained specimens shrunk slightly, and the reduction coefficient was (0.92±0.03), which was reduced by 8% after staining. The entire cutaneous nerve distribution pattern could be seen with the naked eye (Fig 2). In the subcutaneous fat-removed specimens, the small nerve branches in the dermis had diffuse distribution, with some twisted or knotted with each other. The middle 1/3 of the lateral posterior forearm has been used as a representative specimen (Fig 3). The epidermis was exfoliated during the immersion process; hence, no intra-epidermal nerve branches could be seen in the flat specimens.

**LACN.** The anterior branch of lateral antebrachial cutaneous nerve (ABLACN) ran inferomedially through the upper 1/3 of lateral anterior forearm, reaching the middle and lower 1/3 of medial anterior forearm [8–11]. Secondary branches appeared along the way, which were relatively large and constant in the upper 1/3 of anterior forearm. These secondary branches innervated both sides of the midline of the anterior forearm, and its arborized branches coaptated with the branches of posterior branch of lateral antebrachial cutaneous

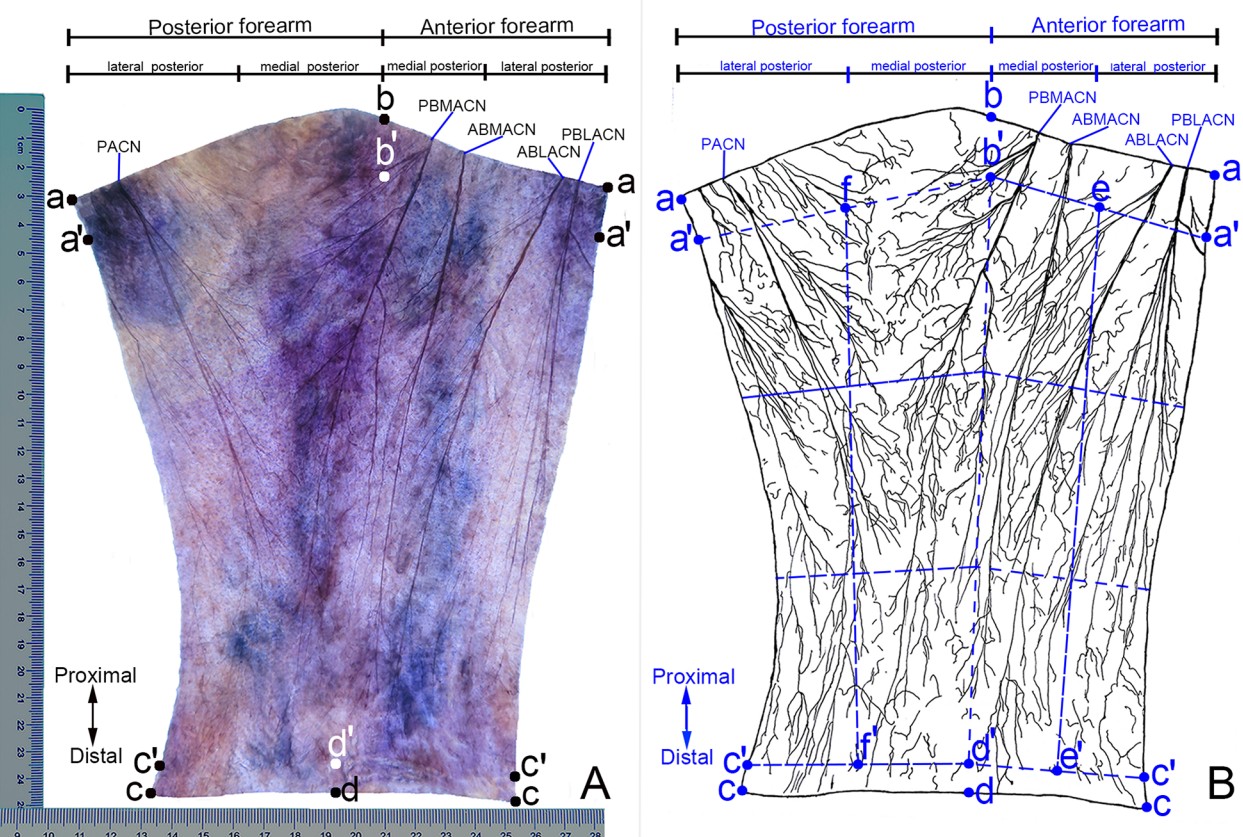

**Fig 2. Distribution pattern of the right forearm cutaneous nerve.** (A) Sihler's staining. (B) Patterns in Figure A. ABLACN: anterior branch of lateral antebrachial cutaneous nerve, PBLACN: posterior branch of lateral antebrachial cutaneous nerve, ABMACN: anterior branch of the medial antebrachial cutaneous nerve, PBMACN: posterior branch of the medial antebrachial cutaneous nerve, PACN: Posterior cutaneous nerve of forearm. Ruler is cm.

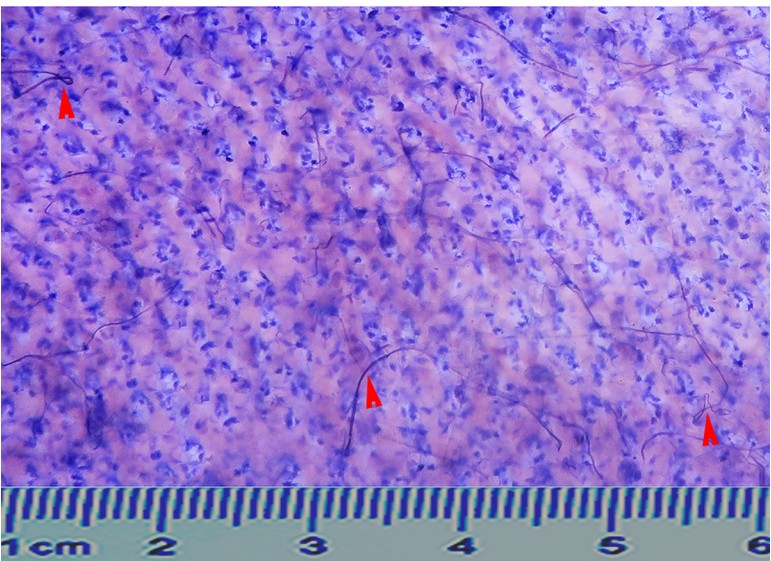

**Fig 3. Sihler's staining revealed the distribution of intradermal nerves in the middle 1/3 area of the lateral posterior region of the right forearm.** Red arrow points to twisted knotted nerve branch. Ruler is cm.

nerve (PBLACN) and anterior branch of medial antebrachial cutaneous nerve (ABMACN). The PBLACN was separated into 4–5 secondary branches in upper 1/3 of the lateral anterior forearm. Among them, the first branch was thicker, lying at the lateral side of the nerve trunk, and turned to the posterior forearm and innervated the lateral edge; it coaptated with the branches of the PACN. The other 3–4 secondary branches traveled to the medial side of the nerve trunk and reached the lower 1/3 of the lateral anterior forearm (Fig 2). The ABLACN innervated (26.00±2.27)% of the medial anterior forearm, whereas the PBLACN innervated (38.21±3.01)% of the lateral anterior forearm and (24.46±2.13)% of the lateral posterior forearm, respectively (Fig 4).

**MACN.** ABMACN projected 7–10 secondary branches. There was one constant and relatively thick secondary branch in the medial upper 1/3 and medial middle 1/3, with a coaptation between the arborized branches of these secondary branches; in the boundary between medial and middle of the anterior forearm, the branches of ABMACN coaptated with the branches of the ABLACN. The middle and lower anterior forearm at the medial border also had a coaptation with the branches of the posterior branch of medial antebrachial cutaneous nerve (PBMACN). Particularly, these branches and coaptations were denser in the middle and lower 1/3 of the medial anterior forearm. The ABMACN mainly distributed to the middle region of the medial anterior forearm, and covered (27.67±2.93)% of anterior forearm (Fig 4A and 4C). After travelling some distance in the upper 1/3 of the medial anterior forearm, the PBMACN turned to the posteromedial border of the forearm, reaching the lower 1/3 of the medial posterior forearm inferolaterally. It sent 10–12 secondary branches along the way, especially in the upper 1/3 of the medial posterior forearm, where they appeared more constant and thicker. Most of these secondary branches turned to the medial posterior forearm and innervated the medial 2/3 of the medial posterior forearm. However, one of these mainly innervated the middle and lower 1/3 of the anterior forearm at the medial border after being separated from the upper 1/3 of the medial forearm. The arborized branches of the secondary branches of the PBMACN coaptated with the branches of ABMACN and PACN (Fig 2). PBMACN covered (7.67±1.71)% of the anterior and (34.75±3.38)% of posterior forearm (Fig 4B and 4D).

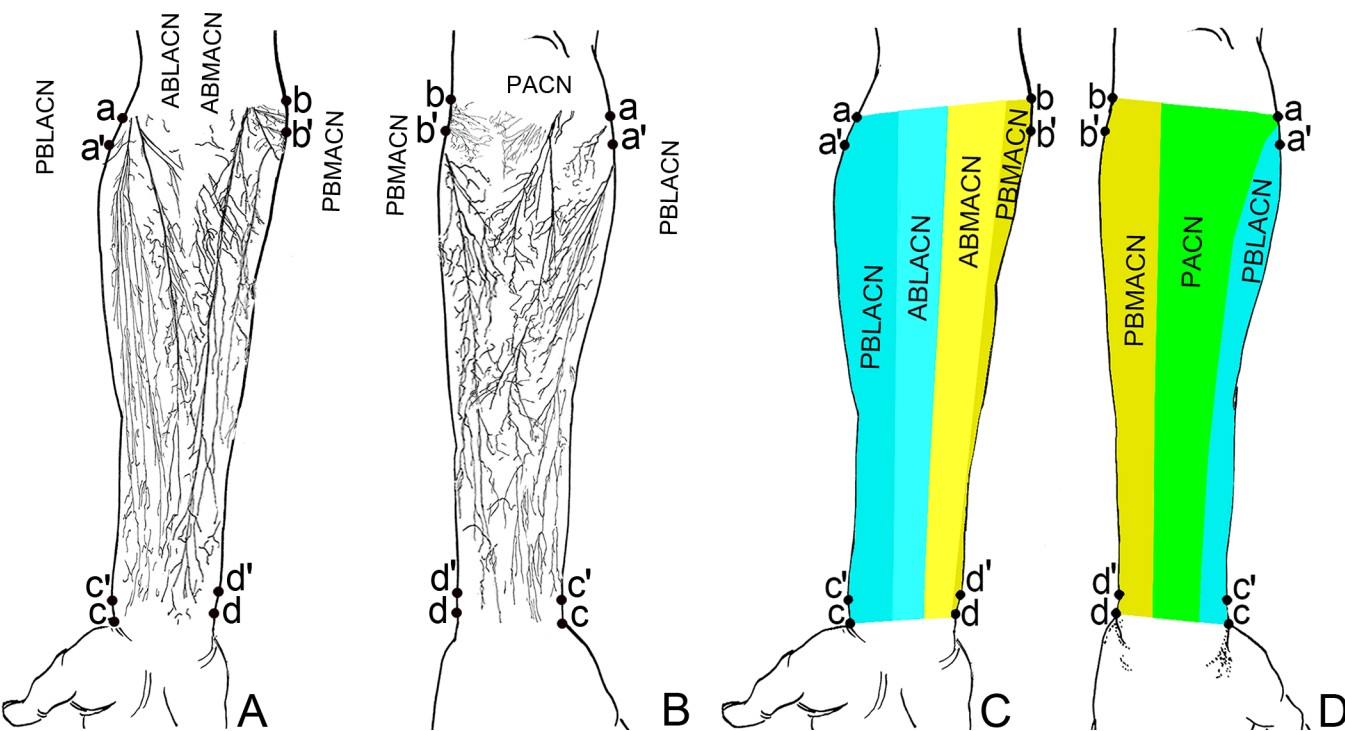

**Fig 4. Distribution pattern and innervation area of the forearm cutaneous nerve.** (A) Sketch map of cutaneous nerve distribution in the anterior forearm. The light blue and light yellow frames represent the ideal donor sites for the design of flaps with ABLACN and ABMACN, respectively. (B) Sketch map of cutaneous nerve distribution in the posterior forearm. The dark blue, dark yellow and the green frames represent the ideal donor site for the design of flaps with PBLACN, PBMACN and PACN, respectively. (C) Area innervated by cutaneous nerves of anterior forearm. (D) Area innervated by cutaneous nerves of posterior forearm.

**PACN.** Beginning at the level of the posterior forearm, the primary branches were in the upper 1/3 of the lateral posterior forearm, with 3–4 branches, especially the first branch, being relatively thick. In one case, the first primary branch of PACN was as thick as the trunk; it can be divided into 2 primary branches: medial and lateral. Most branches of PACN traveled inward, crossing the midline of the posterior forearm to coaptate with the branches of PBMACN, and a few branches of PACN ran outward and communicated with the branches of PBLACN at the junctional zone between the upper and middle 1/3 and that between the middle and lower 1/3 of the lateral posterior forearm (Fig 2). PACN covered (41.04±4.33)% of the posterior forearm (Fig 4B and 4D).

## Thickness, area and nerve branch density of each region

The thickness, area, and nerve branch density, in each forearm region are shown in Table 1. The following was noted: for specimens taken from the muscular surface: thickest region: in the upper 1/3 of medial anterior forearm, thinnest region: in lower 1/3 of lateral anterior forearm, and for fat-removed samples: thickest region: in upper 1/3 of medial posterior forearm, thinnest region: in the lower 1/3 of lateral posterior forearm. The largest area was found in upper 1/3 of the medial posterior forearm, the lower 1/3 of the anterolateral region was the smallest. Comparison of nerve branch densities in different regions of the forearm revealed the relatively dense secondary nerve branches in the middle 1/3 of lateral anterior and medial posterior forearm, and sparse in the lower 1/3 of lateral anterior, upper 1/3 of medial in anterior forearm, lower 1/3 of lateral posterior, upper 1/3 and lower 1/3 of medial posterior forearm.

**Table 1. Comparison of skin thickness, area and density of cutaneous nerve branches in various regions of the forearm.**

| Regions | Thickness of the specimen (cm) | | Skin area (cm²) | | Density of Secondary nerve branches (branch /cm²) | Density of tertiary nerve branches (branch /cm²) | Density of nerve branches in dermis (branch /cm²) | Total nerve branch density (branch/cm²) |
|---|---|---|---|---|---|---|---|---|
| | Subcutaneous fat was not removed | Subcutaneous fat was removed | Before staining | After Staining | | | | |
| Upper 1/3 of anterolateral region | 0.29±0.02 | 0.26±0.01 | 28.82±3.28 | 26.51±3.22 | 0.44±0.11 | 0.24±0.06 | 1.18±0.14 | 1.87±0.24 |
| Middle 1/3 of anterolateral region | 0.25±0.02 | 0.21±0.03 | 22.64±3.88 | 20.83±3.70 | 0.45±0.12 | 0.51±0.12 | 2.15±0.40 | 3.11±0.59 |
| Lower 1/3 of anterolateral region | 0.18±0.02 | 0.18±0.02 | 17.59±3.00 | 16.20±2.91 | 0.26±0.09 | 0.32±0.09 | 1.50±0.30 | 2.07±0.44 |
| Upper 1/3 of anteromedial region | 0.37±0.05 | 0.31±0.05 | 31.09±3.54 | 28.63±3.65 | 0.34±0.07 | 0.46±0.08 | 1.37±0.16 | 2.17±0.26 |
| Middle 1/3 of anteromedial region | 0.27±0.02 | 0.24±0.02 | 26.08±4.12 | 24.02±4.05 | 0.42±0.12 | 0.66±0.17 | 2.48±0.44 | 3.57±0.67 |
| Lower 1/3 of anteromedial region | 0.21±0.02 | 0.19±0.03 | 20.91±2.98 | 19.25±2.93 | 0.38±0.08 | 0.55±0.12 | 2.07±0.33 | 3.00±0.50 |
| Upper 1/3 of posterolateral region | 0.28±0.04 | 0.25±0.02 | 31.64±3.92 | 29.11±3.76 | 0.43±0.07 | 0.39±0.10 | 1.39±0.21 | 2.21±0.33 |
| Middle 1/3 of posterolateral region | 0.26±0.02 | 0.22±0.02 | 21.87±2.80 | 20.15±2.86 | 0.45±0.09 | 0.40±0.10 | 2.72±0.46 | 3.58±0.59 |
| Lower 1/3 of posterolateral region | 0.19±0.02 | 0.15±0.02 | 17.79±2.67 | 16.39±2.68 | 0.28±0.08 | 0.39±0.09 | 2.11±0.32 | 2.77±0.43 |
| Upper 1/3 of posteromedial region | 0.35±0.05 | 0.33±0.05 | 34.09±3.86 | 31.38±3.82 | 0.29±0.07 | 0.34±0.09 | 1.17±0.16 | 1.79±0.26 |
| Middle 1/3 of posteromedial region | 0.33±0.04 | 0.28±0.04 | 28.60±4.81 | 26.34±4.67 | 0.58±0.14 | 0.36±0.10 | 2.14±0.41 | 3.07±0.61 |
| Lower 1/3 of posteromedial region | 0.24±0.03 | 0.17±0.02 | 23.37±3.77 | 21.51±3.67 | 0.30±0.10 | 0.29±0.09 | 2.22±0.40 | 2.81±0.53 |

The relatively dense tertiary nerve branches were found in the middle 1/3 and lower 1/3 of medial anterior forearm, while relatively sparse were found in upper 1/3 and lower 1/3 of lateral anterior, lower 1/3 of medial posterior forearm. The intradermal nerve branches were the relatively dense in the middle 1/3 of medial anterior and lateral posterior forearm, while the upper 1/3 of lateral anterior and medial posterior forearm were sparse. Comparing the total nerve branch density of each region, the middle 1/3 of medial and lateral forearm were the dense, while upper 1/3 and lower 1/3 lateral anterior, the upper 1/3 of medial posterior forearm were sparse. The thickness, area and nerve branches density data comparison showed statistical significance among regions ($P < 0.05$); The dense secondary nerve branches data comparison between the middle 1/3 of the lateral anterior and medial posterior forearm, and the dense tertiary nerve branches data comparison between the middle 1/3 and lower 1/3 of the medial anterior forearm, showed no statistically significant differences ($P > 0.05$). The dense intradermal nerve branch data comparison showed no significant differences between the middle 1/3 of medial anterior and lateral posterior forearm ($P > 0.05$). The density of total nerve branch data comparison showed no significant differences between the middle 1/3 of the medial and lateral forearm ($P > 0.05$). Comparison between the left and the right showed no statistical significance ($P > 0.05$).

## Discussion

Despite the continued efforts of surgeons to improve the patient's quality of life [21], the effect of the flaps designed using the available data on cutaneous nerve distribution is not

satisfactory. Forearm flaps are thin and pliable, and have relatively abundant cutaneous nerves, making them good candidates for repairing defects in the sensitive areas [22]. Our study revealed the overall forearm cutaneous nerve distribution pattern, thus may aid in rationally designing donor-recipient matched flaps for functional restoration and aesthetic reconstruction requirements.

Vascular-neurotrophic flaps are better than traditional flaps, especially considering that cutaneous nerves can restore sensory function. Katou et al. and Netscher et al. used 1/3 radial free flaps of the distal forearm to reconstruct the oral sensory function and found that the innervated flap provided faster and better recovery of sensation than the non-innervated flaps [23, 24]. Another study found no significant difference in the cross-sectional area between LACN and the digital nerve, considering that LACN could be a well-suited donor for digital nerve grafting [10]. Since ABLACN and ABMACN always extend to the distal third of the forearm, the pedicled retrograde-flow vascular-neurotrophic island flaps can be used to repair the distal lateral and dorsal defects of the forearm, thus alleviating the sensory defect of the donor site [16]. Additionally, ulnar flaps of the PACN nutrient vessels can be designed along the midline of the dorsal forearm to repair the defect of the dorsum of one hand, and the maximum area of the flaps can be approximately 5.5×12 cm [25, 26].

These forearm flaps are designed based on the cutaneous nerve trunk and its nutrient vessels, and do not consider the nerve branches and distribution density. For the large branches of the cutaneous nerve of the forearm previously reported, MACN and LACN are usually divided into 2 primary branches [2–3], which is consistent with our results (Figs 2 and 4). We found that PACN has a single trunk (Fig 2), contrary to Maida's conclusion of PACN being usually divided into medial and lateral branches [6]. For small branches, our results suggested that the number of nerve branches was more than that reported by Race et al. through gross anatomy, which suggested that the modified Sihler's nerve staining was superior to gross anatomy [27].

The results of this study suggest that in the forearm free-flap transplantation, when skin flaps innervated by ABMACN need to be transplanted to the recipient site, sensory defects at the donor site can be prevented by maintaining an intact ABMACN (Fig 4A). A thicker secondary nerve branch for the flap design in the upper and middle 1/3 of the medial anterior forearm is recommended. Similarly, when using PBMACN for flap transplantation, it is not necessary to cut the PBMACN, and cutting a thicker secondary nerve branch for the flap design in the upper 1/3 of the posterior forearm on medial side is recommended (Fig 4A and 4B). For ABLACN and PBLACN, the ideal donor sites are in the upper 1/3 of the lateral anterior and posterior forearm (Fig 4A and 4B). For PACN, the flap can be designed with the first primary branch of the upper1/3 of lateral posterior forearm or primary branches of the middle 1/3 of medial posterior forearm (Fig 4B).

Sensory reconstruction primarily occurs as follows: the reception area and surrounding nerve fibers grow into the flap through the scar tissue (peripheral manner), or the recipient nerves are anastomosed with the main innervating nerve of the flap (central manner). The direction of nerve regeneration can be guided by the alignment of nerve tissues, so that Schwann cells at both ends can directly restore material communication channels, and the regenerated nerve fibers can grow smoothly into the distal neurotubules [28]. However, the important factors for flap recovery are distribution of sensory nerves in the donor site of the flap, the choice of the cutaneous nerve, and its site of coaptation [29, 30]. Therefore, if the recipient nerve is rich, a rich donor nerve should be matched to increase the probability of nerve contact. This study showed that the primary, secondary, or tertiary nerve branch was not the densest in the middle 1/3 of the lateral posterior forearm. However, the total nerve branch is the densest, attributable to the fact that some small nerve branches can pass through

muscles to the skin. Thus, this area should be considered as the first choice for transplanting forearm skin flaps to reconstruct sensory defects.

Our study successfully showed the entire distribution pattern, revealing the location of the dense and the areas of coaptation, estimating and depicting the area and extent of the primary branches of the forearm cutaneous nerve, and provided visual information for the selection and matching of materials for sensory reconstruction in flap transplantation.

However, this study has some limitations. The density of intradermal nerve branches may not be as accurate as that obtained using immunohistochemistry and cannot be displayed in the same specimens of intradermal and epidermal nerve branches. Because this study is limited by its samples coming from a single ethnic population, it has not yet revealed whether there are differences among races. Additionally, the dominance of each cutaneous nerve is not exactly consistent with the boundary described in the Netter Atlas and needs to be verified [31].

This study supplemented new information about the forearm cutaneous distribution pattern and provides morphological guidance to clinicians, which can help design flaps for transplantations to repair the sensory function of the recipient site, with considerations of matching the donor and recipient graft site sensory nerve fibers.

## Acknowledgments

The authors would like to acknowledge Yunqiang Zhang for his technical assistance during photography. We also would like to thank the silent teachers for their selfless dedication.

## Author Contributions

**Conceptualization:** Zairong Wei, Shengbo Yang.

**Data curation:** Hui Li.

**Methodology:** Shengbo Yang.

**Writing – original draft:** Hui Li, Weiwei Zhu, Shouwen Wu.

**Writing – review & editing:** Zairong Wei, Shengbo Yang.

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
