## [Decision Letter · Decision Letter 0]

10 Jul 2019

PONE-D-19-16142

Anatomical Analysis of Antebrachial Cutaneous Nerve Distribution Pattern and Its Clinical Implications for Sensory Reconstruction

PLOS ONE

Dear Dr Yang,

Thank you for submitting your manuscript to PLOS ONE. After careful consideration, we feel that it has merit but does not fully meet PLOS ONE’s publication criteria as it currently stands. Therefore, we invite you to submit a revised version of the manuscript that addresses the points raised during the review process.

We would appreciate receiving your revised manuscript by Aug 24 2019 11:59PM. To enhance the reproducibility of your results, we recommend that if applicable you deposit your laboratory protocols in protocols.io, where a protocol can be assigned its own identifier (DOI) such that it can be cited independently in the future. For instructions see: http://journals.plos.org/plosone/s/submission-guidelines#loc-laboratory-protocols

We look forward to receiving your revised manuscript.

Kind regards,

Leila Harhaus

Academic Editor

PLOS ONE

**Journal Requirements**

2. Thank you for including your ethics statement:  "Twelve adult cadavers, 8 men and 4 women.The use of these specimens was authorized by the ethics committee of our school.".   

a.Please amend your current ethics statement to include the full name of the ethics committee/institutional review board(s) that approved your specific study.

b.Please amend your current ethics statement to confirm that your named institutional review board or ethics committee specifically approved this study.

c.Once you have amended this/these statement(s) in the Methods section of the manuscript, please add the same text to the “Ethics Statement” field of the submission form (via “Edit Submission”).

3. Thank you for stating that “The funders had no role in study design, data collection and analysis, decision to publish, or preparation of the manuscript” in your financial disclosure.

Please also provide the name of the funders of this study (as well as grant numbers if available) in your financial disclosure statement.

**Comments to the Author**

1. Is the manuscript technically sound, and do the data support the conclusions?

Reviewer #1: Yes

Reviewer #2: Yes

Reviewer #3: Yes

2. Has the statistical analysis been performed appropriately and rigorously? 

Reviewer #1: N/A

Reviewer #2: Yes

Reviewer #3: I Don't Know

3. Have the authors made all data underlying the findings in their manuscript fully available?

Reviewer #1: Yes

Reviewer #2: Yes

Reviewer #3: Yes

4. Is the manuscript presented in an intelligible fashion and written in standard English?

Reviewer #1: Yes

Reviewer #2: Yes

Reviewer #3: Yes

5. Review Comments to the Author

Reviewer #1: The paper is original and adds to the knowledge base in this area using a novel approach to staining and defining the branching anatomy. The paper is well written and no substantial revisions are required. In line 73 the “A” should be non capital and in line 173 there should be a space between “Fig” and “3”.

Reviewer #2: DEAR AUTHOR,

THIS STUDY IS A VERY GOOD CADAVERIC WORK, DISPLAYING FOREARM SENSITIVE DISTRUBUTION WITH SOME LIMITATIONS CONCERNING RACE DIFFERENCES. HOWEVER CLINICAL USE OF THIS KNOWLEDGE ( YOUR FINDINGS ) IS DEBATABLE SINCE, FLAP ELEVATION STILL DEPENDS ON VASCULAR (PERFORATOR VESSEL) DISTRIBUTION.

I HAVE SOME DOUBTS REGARDING DONOR AREA SENSITIVE NERVE THICKNESS IS DIRECTLY CORRELATED WITH BETTER SENSITIVE OUTCOME, YOU WOULD RATHER SMOOTHEN YOUR IDEAS OR SUPPORT WTH MORE DATA.

Reviewer #3: This study describes distribution of forearm cutaneous nerves including gross anatomy, main and minor branches, interconnections and density. One of the aims is to give clinical suggestions to improve flap design in a matter or sensory reconstruction at recipient site and possible sensory loss at donor site.

The study is well designed and appropriate methods were used including application of modified Sihler's staining to reveal and analyze cutaneous distribution.

Advantages:

- novel data about branches distribution, density and interconnections between major antebrachial nerves

Disadvantages:

- the process of pictures conversion and further image processing is not well described

- although authors describe the results thickness, area and nerve branch density of each region (p.10) they provide only general ANOVA result (p. 11) without post-hoc analysis

- clinical implications are hard to drawn due to big amount of data in multiple regions - most suitable donor sites could be marked on separate figure

In the summary the study gives relevant and precise anatomical information but clinical implications are not so clear to me at this moment.

6. PLOS authors have the option to publish the peer review history of their article (what does this mean?). If published, this will include your full peer review and any attached files.

Reviewer #1: No

Reviewer #2: No

Reviewer #3: Yes: Piotr Czarnecki

---

## [Author Response · Author response to Decision Letter 0]

25 Jul 2019

Jul 25, 2019

Academic Editor,

PLoS One

PONE-D-19-16142

Dear Editor:

Thank you for your letter on Jul 25, 2019 regarding this manuscript. We would also like to thank the reviewers for their helpful comments. We are now submitting a revised version as detailed below that addresses these comments (original comments from reviewers and the editor are in italics). All changes have been indicated in red in the revised manuscript.

Response to Reviewer#1:

The paper is original and adds to the knowledge base in this area using a novel approach to staining and defining the branching anatomy. The paper is well written and no substantial revisions are required. In line 73 the “A” should be non capital and in line 173 there should be a space between “Fig” and “3”

Response:

Thank you for your positive comments. As for the two errors you pointed out, we have revised the text accordingly.

Response to Reviewer #2:

1. THIS STUDY IS A VERY GOOD CADAVERIC WORK, DISPLAYING FOREARM SENSITIVE DISTRUBUTION WITH SOME LIMITATIONS CONCERNING RACE DIFFERENCES.

Response:

(1) Thank you for your positive comment. 

(2) Because of the restrictions specimen resource, Our study could not answer racial differences indeed. 

(3) However, we did the following literature search about the current understanding of the “SENSITIVE DISTRUBUTION” in different races:

① Antebrachial Cutaneous Nerve trunk and its main branches’ distribution have been reported by authors in American, China, Austria, and Korea [1-8]. The nerve distributions described in these studies are essentially consistent. There are variations between individual samples but none mentioned racial differences. 

② Studies by Lauria et al show that there are racial differences in cutaneous innervation of the leg among European, American and Asian populations [9]. However, Collongues et al. found that no racial difference showed in clinical correlations [10].

③ Tenny et al [11] in 1984 used the lateral antebrachial cutaneous nerve graft for traumatic digital nerve defects where they found that “age” was a more important prognostic factor for sensibility. However, The study did not mention racial differences.

④ Our study is about the overall display and comparison of the forearm nerve distribution density in each region. As far as we know, neurosensitivity cannot be equated with nerve density, and sensitivity may be affected by gender, age, personal living conditions, etc.

In general, we hope that further research in the future will confirm whether there are racial differences in the forearm nerve distribution.

1. Finneran JJ, Sandhu N. Ultrasound-Guided Posterior Antebrachial Cutaneous Nerve Block: Technical Description and Block Distribution in Healthy Volunteers. J Ultrasound Med. 2019; 38: 239-242. doi: 10.1002/jum.14678. PMID: 29732596.

2. Singhal S, Rao VV, Ravindranath R. Variations in brachial plexus and the relationship of median nerve with the axillary artery: A case report. J Brachial Plex Peripher Nerve Inj. 2007; 2: 21. doi: 10.1186/1749-7221-2-21. PMID: 17915015.

3. Race CM, Saldana MJ. Anatomic course of the medial cutaneous nerves of the arm. J Hand Surg Am. 1991; 16: 48–52. PMID: 1995693.

4. Zhang FH, Topp SG, Zhang WJ, Zheng HP, Zhang F. Anatomic study of distally based pedicle compound flaps with nutrient vessels of the cutaneousnerves and superficial veins of the forearm. Microsurgery. 2006; 26: 373-85. doi: 10.1002/micr.20255. PMID: 16783807.

5. Benedikt S, Parvizi D, Feigl G, Koch H. Anatomy of the medial antebrachial cutaneous nerve and its significance in ulnar nerve surgery: An anatomical study. J Plast Reconstr Aesthet Surg. 2017 Nov; 70: 1582-1588. doi: 10.1016/j.bjps.2017.06.025. PMID: 28756975.

6. Moritz T, Prosch H, Pivec CH, et al. High-resolution ultrasound visualization of the subcutaneous nerves of the forearm: A feasibility study in anatomic specimens. Muscle Nerve. 2014; 49: 676-679. doi: 10.1002/mus.24064. PMID: 24038104.

7. Im HS, Im JY, Kim KH,Kim DH, Park BK. Ultrasonographic Study of the Anatomical Relationship Between the Lateral Antebrachial Cutaneous Nerve and the Cephalic Vein. Ann Rehabil Med. 2017; 41: 421-425. doi: 10.5535/arm.2017.41.3.421. PMID: 28758079.

8. Oh CH, Park NS, Kim JM, Kim MW. Determination of an ideal stimulation site of the medial antebrachial cutaneous nerve using ultrasound and investigation of the efficiency. Ann Rehabil Med. 2014; 38: 836–842. doi: 10.5535/arm.2014.38.6.836. PMID: 25566484.

9. Lauria G, Bakkers M, Schmitz C, Lombardi R, Penza P, Devigili G, et al. Intraepidermal nerve fiber density at the distal leg: a worldwide normative reference study. J Peripher Nerv Syst. 2010; 15: 202–207. doi: 10.1111/j.1529-8027.2010.00271.x. PMID: 21040142.

10. Collongues N, Samama B, Schmidt-Mutter C, Chamard-Witkowski L, Debouverie M, Chanson JB. et al. Quantitative and qualitative normativedataset for intraepidermal nerve fibers usingskin biopsy. PLOS ONE. 2018; 13: e0191614. doi: 10.1371/journal.pone.0191614. PMID: 29370274.

11. Tenny JR, Lewis RC. Digital nerve-grafting for traumatic defects. Use of the lateral antebrachial cutaneous nerve.JBone Joint Surg Am. 1984; 66: 1375–1379. PMID: 6501333.

2. CLINICAL USE OF THIS KNOWLEDGE ( YOUR FINDINGS ) IS DEBATABLE SINCE, FLAP ELEVATION STILL DEPENDS ON VASCULAR (PERFORATOR VESSEL) DISTRIBUTION.

I HAVE SOME DOUBTS REGARDING DONOR AREA SENSITIVE NERVE THICKNESS IS DIRECTLY CORRELATED WITH BETTER SENSITIVE OUTCOME, YOU WOULD RATHER SMOOTHEN YOUR IDEAS OR SUPPORT WTH MORE DATA.

Response:

We completely agree with you: “FLAP ELEVATION STILL DEPENDS ON VASCULAR (PERFORATOR VESSEL) DISTRIBUTION”. Our study aimed at improving the quality of the flaps by considering the recipient sensory recovery so that the clinicians can use this knowledge to guide the design of the flaps. We have also expressed these views in " Introduction, Paragraph 2 ".

The theoretical design of this study is based on references 28-30 listed in this paper. Please also refer to the “Discussion section, paragraph 5”. “Sensory reconstruction primarily occurs as follows: the reception area and surrounding nerve fibers grow into the flap through the scar tissue (peripheral manner), or the recipient nerves are anastomosed with the main innervating nerve of the flap (central manner). The direction of nerve regeneration can be guided by the alignment of nerve tissues, so that Schwann cells at both ends can directly restore material communication channels, and the regenerated nerve fibers can grow smoothly into the distal neurotubules [28]. However, the important factors for flap recovery are distribution of sensory nerves in the donor site of the flap, the choice of the cutaneous nerve, and its anastomotic site [29, 30]. Therefore, if the recipient nerve is rich, a rich donor nerve should be matched to increase the probability of nerve contact.” We hope that this reviewer will accept our views. 

28. Ide C, Tohyama K, Yokota R, Nitatori T, Onodera S. Schwann cell basal lamina and nerve regeneration. Brain Res. 1983; 288: 61-75. doi: 10.1016/0006-8993(83)90081-1. PMID: 6661636.

29. Feng SM, Wang AG, Zhang ZY, Sun QQ, Tao YL, Zhou MM, et al. Repair and sensory reconstruction of the children's finger pulp defects with perforator pedicled propeller flap in proper digital artery. Eur Rev Med Pharmacol Sci. 2017; 21: 3533-3537. PMID: 28925493.

30. Zhu L, Zhang J, Song X, Hou W, Wu S, Chen W, et al. Sensory recovery of non-innervated free flaps and nasolabial island flaps used for tongue reconstruction of oncological defects. J Oral Rehabil. 2017; 44: 736-748. doi: 10.1111/joor.12510. PMID: 28370156.

Response to Reviewer #3:

1. Advantages：omit.

2. Disadvantages:

- the process of pictures conversion and further image processing is not well described.

- although authors describe the results thickness, area and nerve branch density of each region (p.10) they provide only general ANOVA result (p. 11) without post-hoc analysis

- clinical implications are hard to drawn due to big amount of data in multiple regions - most suitable donor sites could be marked on separate figure.

Response:

In order to better intuitively understand the ideal donor sites, we have marked in Figs 4A and 4B. We did not add new figures for the concern of reducing the pages of the manuscript.

About the clinical significance of this study: We analyzed the clinical application of three cutaneous nerves in the forearm based on the experimental results. These descriptions are in the fourth paragraph of the discussion section.

We added post-hoc analysis in statistical processing (see revised text).

Response to Editor

Thank you for your consideration of this revision. The editing of the text is based on the template of your journal.

Sincerely,

Shengbo Yang

Department of Anatomy

Zunyi Medical University

---

## [Decision Letter · Decision Letter 1]

22 Aug 2019

PONE-D-19-16142R1

Anatomical Analysis of Antebrachial Cutaneous Nerve Distribution Pattern and Its Clinical Implications for Sensory Reconstruction

PLOS ONE

Dear Dr Yang,

Thank you for submitting your manuscript to PLOS ONE. After careful consideration, we feel that it has merit but does not fully meet PLOS ONE’s publication criteria as it currently stands. Therefore, we invite you to submit a revised version of the manuscript that addresses the points raised during the review process.

We would appreciate receiving your revised manuscript by Oct 06 2019 11:59PM. To enhance the reproducibility of your results, we recommend that if applicable you deposit your laboratory protocols in protocols.io, where a protocol can be assigned its own identifier (DOI) such that it can be cited independently in the future. For instructions see: http://journals.plos.org/plosone/s/submission-guidelines#loc-laboratory-protocols

We look forward to receiving your revised manuscript.

Kind regards,

Leila Harhaus

Academic Editor

PLOS ONE

Reviewers' comments:

Reviewer's Responses to Questions

**Comments to the Author**

1. If the authors have adequately addressed your comments raised in a previous round of review and you feel that this manuscript is now acceptable for publication, you may indicate that here to bypass the “Comments to the Author” section, enter your conflict of interest statement in the “Confidential to Editor” section, and submit your "Accept" recommendation.

Reviewer #2: All comments have been addressed

Reviewer #3: All comments have been addressed

2. Is the manuscript technically sound, and do the data support the conclusions?

Reviewer #2: Yes

Reviewer #3: Yes

3. Has the statistical analysis been performed appropriately and rigorously? 

Reviewer #2: Yes

Reviewer #3: Yes

4. Have the authors made all data underlying the findings in their manuscript fully available?

Reviewer #2: Yes

Reviewer #3: Yes

5. Is the manuscript presented in an intelligible fashion and written in standard English?

Reviewer #2: Yes

Reviewer #3: Yes

6. Review Comments to the Author

Reviewer #2: Forearm sensitive nerve distrubution is important in terms of donor area selection and flap choices so I think your manuscript will be usefull for the scientists dealing with this subject.However terminology is important as well so all "anastomosis" words must be replaced with coaptation or neuroraphy.

Reviewer #3: (No Response)

7. PLOS authors have the option to publish the peer review history of their article (what does this mean?). If published, this will include your full peer review and any attached files.

Reviewer #2: Yes: ATAKAN AYDIN

Reviewer #3: Yes: Piotr Czarnecki

---

## [Author Response · Author response to Decision Letter 1]

25 Aug 2019

Aug 25, 2019

Academic Editor,

PLOS ONE

PONE-D-19-16142R1

Dear Editor:

Thank you for your letter on Aug 23, 2019 regarding this manuscript. We would also like to thank the reviewers for their helpful comments. We are now submitting a revised version as detailed below that addresses these comments (original comments from reviewers and the editor are in italics). All changes have been indicated in red in the revised manuscript.

Response to Reviewer#2:

Forearm sensitive nerve distrubution is important in terms of donor area selection and flap choices so I think your manuscript will be usefull for the scientists dealing with this subject. However terminology is important as well so all "anastomosis" words must be replaced with coaptation or neuroraphy.

Response:

Thank you for your positive comment. According to your comments, we have replaced "anastomosis" with coaptation.

Response to Editor:

We have revised the text in accordance with the comments of the reviewers. 

Sincerely,

Shengbo Yang

Department of Anatomy

Zunyi Medical University

---

## [Editor Report · Decision Letter 2]

28 Aug 2019

Anatomical Analysis of Antebrachial Cutaneous Nerve Distribution Pattern and Its Clinical Implications for Sensory Reconstruction

PONE-D-19-16142R2

Dear Dr. Yang,

We are pleased to inform you that your manuscript has been judged scientifically suitable for publication and will be formally accepted for publication once it complies with all outstanding technical requirements.

With kind regards,

Leila Harhaus

Academic Editor

PLOS ONE

---

## [Editor Report · Acceptance letter]

4 Sep 2019

PONE-D-19-16142R2 

Anatomical Analysis of Antebrachial Cutaneous Nerve Distribution Pattern and Its Clinical Implications for Sensory Reconstruction 

Dear Dr. Yang:

I am pleased to inform you that your manuscript has been deemed suitable for publication in PLOS ONE. Congratulations! Your manuscript is now with our production department. 

With kind regards,

on behalf of

Prof. Dr. med. Leila Harhaus 

Academic Editor

PLOS ONE